# Enrichment of Water Bodies with Phenolic Compounds Released from *Betula* and *Pinus* Pollen in Surface Water

**DOI:** 10.3390/plants13010099

**Published:** 2023-12-28

**Authors:** Ilona Kerienė, Ingrida Šaulienė, Laura Šukienė, Asta Judžentienė, Magdalena Ligor, Gintaras Valiuškevičius, Dalia Grendaitė, Bogusław Buszewski

**Affiliations:** 1Regional Development Institute, Šiauliai Academy, Vilnius University, 84 Vytauto Str., LT-76352 Šiauliai, Lithuania; ingrida.sauliene@sa.vu.lt (I.Š.); laura.sukiene@sa.vu.lt (L.Š.); 2Department of Organic Chemistry, Center for Physical Sciences and Technology, Saulėtekio Avenue 3, LT-10257 Vilnius, Lithuania; asta.judzentiene@ftmc.lt; 3Department of Environmental Chemistry and Bioanalytics, Faculty of Chemistry, Nicolaus Copernicus University, 7 Gagarina Str., 87-100 Torun, Poland; magdalena.ligor@umk.pl (M.L.); bbusz@umk.pl (B.B.); 4Department of Hydrology and Climatology, Faculty of Chemistry and Geosciences, Vilnius University, M. K. Čiurlionio Str. 21, LT-03101 Vilnius, Lithuania; gintaras.valiuskevicius@gf.vu.lt (G.V.); dalia.grendaite@chgf.vu.lt (D.G.)

**Keywords:** *Betula*, *Pinus*, pollen, water bodies, trophicity, bioactivity, phenolic acids, flavonoids

## Abstract

*Betula* and *Pinus* pollen, which are dispersed in natural surface waters, release biologically active compounds into the water bodies. This study aims to evaluate variations in the distribution and composition of phenolic compounds in suspended particles in natural water bodies during pollen spreading. Samples taken from water bodies of different trophic levels were analyzed by microscopy, UV/VIS spectroscopy, HPTLC, and HPLC/DAD. The study revealed that the total phenolic content in water-suspended particles varied from 3.0 mg/g to 11.0 mg/g during *Betula* and *Pinus* pollen spreading. It was also observed that the surface water of dystrophic natural lakes had a higher content of phenolic compounds than the eutrophic, hypereutrophic, and mesotrophic water bodies. Chlorogenic, trans-ferulic, vanillin, and 3,4-dihydroxybenzoic acids were frequently detected in the surface water samples. Experimental measurements have shown variations in the release of phenolic compounds from *Betula* pollen into water (*p* < 0.05). After the exhibition of pollen, the distilled water predominantly contained bioactive chlorogenic acid. Further in situ investigations are necessary to gain a more comprehensive understanding of the function of phenolic compounds in aquatic ecosystems. The exploration of the release of bioactive compounds from pollen could provide valuable insights into the potential nutritional value of pollen as a nutrient source for aquaculture.

## 1. Introduction

*Betula* and *Pinus* plants release large quantities of pollen in May–June, which can be transported over long distances through the air [1]. The pollen production of *Betula* can range from 4.8 to 8.2 million pollen per catkin, depending on the intensity of the shedding season [2]. Similarly, *Pinus* forests generate substantial amounts of *Pinus* pollen biomass, ranging from 100 to 1000 kg/ha. Part of the pollen dispersed by the plants may end up in open water bodies, where it can be deposited in lakes, contributing to a pollen load of 1.47–7.35 g/m^2^ per year [3]. The deposition of *Pinus* pollen in the Baltic Sea has been confirmed using multi-sensor satellite technology [4]. The release of airborne pollen from plants enriches nearby water bodies with nutrients and significantly contributes organic carbon to them [5]. Pollen mainly consists of carbohydrates, proteins, and sporopollenin, along with other constituents [6]. Substances released from pollen can originate from both the outer surface and the interior of the pollen grains. When it comes into contact with water, pollen releases certain substances from its surface [7,8]. Alternatively, depending on the size and elasticity of the pollen pores, they can rupture and disperse the cytoplasmic contents [9]. Pollen decomposition in water bodies involves chemical leaching first, followed by microbial degradation [5]. Phenolic compounds form a part of the pollen chemical constituents [6,10,11,12]. Comprehensive research has explored the bioactive properties, composition, and physiological/ecological aspects of phenolic compounds in terrestrial plants [3,13,14,15,16], revealing their accumulation in aquatic plants [17] and microorganisms [18]. Understanding pollen phenolic compounds is important for studies on plant biodiversity, sexual reproduction, plant–pollinator interactions, aeroallergen monitoring, and climate and pollution effects [3,6,14]. The application of bioactive compounds found in *Betula* and *Pinus* pollen for pharmacological purposes is of limited use, although individual parts of the plants are widely used for health treatments [19,20]. Similarly, there is limited knowledge regarding the release of phenolic compounds into the environment from naturally occurring pollen in ecosystems. In previous research [21], we showed that the content of free and bound phenolic compounds (PC) differed in pollen extracts, with a 20% higher amount of free PC in *Betula* pollen compared to *Pinus* pollen extracts.

A study of the distribution of phenolic compounds released from *Betula* and *Pinus* pollen in water bodies of different morphometric parameters in the territory of Lithuania is presented in the present article. Lithuania is located in a moderate climate zone, which is characterized by an excess of humidity: the average annual temperature in the country is 7.4 °C, and the average annual rainfall is 695 mm [22]. Plains prevail in the country, and most of the area is occupied by agricultural landscapes located in a zone of mixed forests. There are about 2700 lakes and 1150 ponds larger than 0.5 ha in Lithuania (standing water bodies occupy about 2% of the country’s territory) [23,24,25]. The studied lakes and ponds are small and shallow: the surface area varies from 0.035 to 0.665 km^2^, and the average depth does not exceed 5 m (classified as shallow and moderately deep lakes). Such water bodies are very typical for Lithuania: more than 90% of the country’s lakes and ponds are characterized by similar indicators [23]. Like most of the larger lakes in Lithuania, all natural lakes analyzed in the study were formed 10–13 thousand years ago [26]. Most of the artificial inland water bodies in the country are significantly younger (predominately dammed reservoirs and excavated ponds) [27]. The artificial water bodies examined in this study are also like this: Damba fishpond was formed in the 19th century, and the exploited part of the Kalniškiai gravel quarry was flooded with water at the beginning of the 21st century. All studied water bodies are open: small streams flow from them (therefore, they are characterized by a relatively rapid water turnover). Most of the shallow lakes and ponds of Lithuania are eutrophic [28]. The water bodies selected for the study are also affected by natural and anthropogenic eutrophication. The Baltic Sea is an inner sea of the Atlantic Ocean. The current contours of the Baltic Sea were formed between 13,000 and 12,000 years ago. The Lithuanian coast is bordered by the central part of the Baltic Sea, which is shallow and characterized by low (6–8‰) water salinity [29]. Large quantities of biogenic substances brought from the basins in the Baltic Sea and the Curonian Lagoon cause active anthropogenic eutrophication.

The complexity of aquatic ecosystems is determined by a variety of interacting factors. There is still a limited understanding of the contribution of plant phenolic compounds to aquatic ecosystems. It is expected that pollen deposition on surface water bodies during the flowering season may be a source of phenolic compounds to the water bodies. We aimed to understand the distribution and composition of phenolic compounds in water-suspended particles and surface water by analyzing water samples from various trophic statuses of water bodies during the *Betula* and *Pinus* pollen-spreading seasons. It is hypothesized that the release of phenolic compounds from pollen to water differs between the flowering of *Betula* and *Pinus*.

This study provides new insights into the contribution of pollen from anemophilous plants, such as *Betula* and *Pinus*, to the composition of the suspended matter. The study extends the understanding of pollen persistence on water surfaces and the characteristics that indicate the specificity of phenolic compound release from plants. This knowledge can be beneficial for future studies such as modeling the nutritional potential of aquatic ecosystems or analyzing the transformation of plants’ phenolic compounds in aquatic environments. The research findings are valuable for practical applications, as they can be used to address the issue of improving the nutritional environment of aquatic biota.

## 2. Results

### 2.1. Pollen-Spreading Intensity and Pollen Coverage of the Water Surface during the Flowering of Betula and Pinus

The intensity of *Betula* and *Pinus* pollen spreading in Lithuania was analyzed according to the data from the aerobiological monitoring stations in Šiauliai and Klaipėda. Water-suspended particles and surface water samples were collected from the lakes situated in Kurtuvėnai Regional Park, around 20 km away from the aerobiological monitoring station in Šiauliai. The marine water-suspended particles and surface water sampling location was in Klaipėda city; and Curonian lagoon sampling took place in the Klaipėda region (approximately 15 and 30 km away from the aerobiological monitoring station in Klaipėda). Previous studies [30] have led us to assume that *Betula* and *Pinus* pollen concentrations in the study areas are comparable to those measured at the monitoring stations.

According to the data obtained from the aerobiological monitoring stations, the intensity of pollen spreading was high during sampling period. In Klaipeda, the daily concentration of *Betula* pollen varied from 88 to 193 pollen/m^3^, and *Pinus* varied from 63 to 372 pollen/m^3^. In Šiauliai, the daily concentration of *Betula* pollen varied from 217 to 1738 pollen/m^3^, and *Pinus* varied from 208 to 397 pollen/m^3^.

Satellite images confirmed the spread of pollen on the surface of water bodies during the sampling periods (Figure 1). The data from PlanetScope Super Dove satellites were analyzed to determine the extent of pollen coverage on the surface water of water bodies during the *Betula* and *Pinus* plants’ flowering periods. The pollen slicks in the Baltic Sea near Giruliai (BSC) in PlanetScope (Figure 1a) and Sentinel-2 (Figure 1b) satellite images were observed. The higher spatial resolution of PlanetScope satellites facilitated the identification of precise pollen distribution patterns (Figure 1a). We identified pollen as brighter image slicks that show a specific spectral signature (Figure 1c,d). The brightest pixels of image slicks with pollen showed a lower reflectance in the blue band and relatively increased reflectance at green to near infrared wavelengths. The most prominent feature in the images on the PlanetScope image (Figure 1c) was located around point 1 and may indicate the same pollen slick present at points 4 and 5 on the Sentinel-2 image (Figure 1d). Sentinel-2 satellites have a spatial resolution of 10 m, and PlanetScope satellites have a resolution of 3.7 m. Moreover, the PlanetScope satellite passed the location earlier in the day and captured cloudless views of the area, whereas the Sentinel-2 image was taken an hour later, and some pollen slicks were covered by clouds.

### 2.2. Pollen Content and Composition in Water-Suspended Particles

Samples of water-suspended particles were collected during *Betula* (indicated as SPB) and *Pinus* (indicated as SPP) flowering. Sampling was performed in two stages, from water bodies with different trophic statuses, targeting areas abundant in *Betula* and/or *Pinus* plants. Additionally, surface water samples (respectively, SPBW and SPPW) were collected at the same time as water-suspended particles during *Betula* (SPB) and *Pinus* pollen-spreading seasons (SPP) (Table 1).

Throughout *Betula* pollen release and SPB and SPBW sampling periods, there was little rainfall (Appendix A), and no detectable *Betula* pollen was found on water surfaces. The highest concentration of pollen in suspended particles (about 50%) was found in the samples collected at DNLJ (Table 1). The flooded quarry (FQK) and dammed natural lake (DNLB) had a range of 21% to 10% of pollen content in SPB, and other water bodies had 6% to 3% of pollen content. The pollen diversity was mainly composed of *Betula*, accounting for 83–98% of the total (Table 1). Samples had pollen from *Corylus*, *Alnus*, *Fraxinus*, and *Carpinus*. Between 29 April and 10 May 2022, *Pinus* plants were in the early stages of flowering, and their pollen content in SPB ranged from 1 to 10%.

Samples of SPP and SPPW were collected during the *Pinus* pollen-spreading period after rainfall (Appendix A). *Pinus* pollen agglomerates covered the surface of water bodies along the coasts (Figure 1a and Appendix A). The SPP samples exposed a *Pinus* pollen content ranging from 80% to 95%, with a low percentage of other particles. The highest concentration of non-pollen particles was observed in the NLS water body (50%). *Picea* plants grew near all the water bodies. However, pollen content in SPP was low, ranging from 0.2 to 2.0%. *Picea* does not appear to have shed much pollen in 2022, as the air samples collected in Šiauliai and Klaipėda showed low levels of pollen: on average, ~1–2 pollen/m^3^/day.

Concerning water bodies’ trophicity, the dystrophic natural lake (NLS) exposed the lowest pollen content in both SPB and SPP. On the other hand, the eutrophic-hypertrophic dammed natural lake DNLJ had the highest amount of pollen in samples, mostly from *Betula* and *Pinus* plants.

### 2.3. Total Phenolic Content in Water-Suspended Particles and Surface Water during Betula and Pinus Pollen-Spreading Period

The phenolic compounds present in plant cell walls (bound PC) and those existing as free phenolic compounds (free PC) were analyzed in extracts of SPB and SPP. During the *Betula* pollen-spreading period, the content of bound PC in SPB extracts (Table 2) ranged from 3.0 mg/g in a mesotrophic-eutrophic natural lake (DNLB) to 10.9 mg/g in a mesotrophic artificial (FQK) water body.

Phenolic compounds were low in the marine eutrophic water samples (BSC). The SPP extracts from the BCS water body contained sand impurities, which probably increased the dry weight. A similar total phenolic content (TPC) in SPB extracts (9.1 mg/g) was found in hypertrophic water bodies (CLC, DFD). However, there was no correlation between total phenolic and pollen content in SPB extracts (rs = 0.31, *p* = 0.5, *n* = 16). The free PC total phenolic content in SPB extracts ranged from 1.0 mg/g (BSC) to 4.2 mg/g (DFD) and was lower than the bound PC content found in the same water bodies (Table 2).

During the *Pinus* pollen-spreading period, the total phenolic content in SPP extracts of bound PC and free PC was similar (Table 2). More phenolic compounds were found in the mesotrophic-eutrophic natural lake (DNLB, ~11.0 mg/g) and in the eutrophic-hypereutrophic natural lake (DNLJ, ~10 mg/g). During SPP sampling, some *Pinus* pollen was in the decomposition phase in the hypereutrophic (CLC, DFD) and mesotrophic (FQK) water bodies. Therefore, it could have caused the low total phenolic compounds in the SPP (Table 2).

The results of surface water extracts (PW, SPBW, SPPW) showed that phenolic compounds were present in the water throughout the whole study period (Figure 2).

Before the pollen-spreading period (PW extracts), the dystrophic (NLS) and eutrophic with indications of dystrophy (NLG) water bodies had a higher concentration of phenolic compounds than the others—respectively, 1.24 mg/mL and 1.78 mg/mL (Figure 2). During the *Betula* spreading period, the higher content of phenolic compounds was in the SPBW surface water of mesotrophic (FQK and DNLB) water bodies (Figure 2). Other aquatic plants in the FQK likely influence phenolic compound content (Appendix A).

The differences between the PW and SPBW surface water data show that the content of phenolic compounds increased from 34% (CLC) to 90% (FQK) in eutrophic and mesotrophic water bodies during the *Betula* pollen-spreading period. Thereafter, phenolic content became similar to that in SPPW surface water. A trend was evident that the surface water of dystrophic natural lakes (NLS) had higher phenolic content than that of the other water bodies (Figure 2).

Antioxidant activity according to ABTS• and DPPH• binding intensity in water-suspended particles and surface water was low or below the limit of method detection. Relatively more bioactive compounds were in SPB: 9–13%, according to the ABTS• radical-scavenging activity (Appendix A). A trend was observed that during the *Betula* pollen-spreading period, water-suspended particles and surface water in hypereutrophic water bodies showed a more bioactive background compared to the *Pinus* pollen-spreading period. Additionally, the antioxidant activity of water-suspended particles and surface water in the dystrophic NLS was higher than that in other water bodies (Appendix A).

### 2.4. Analysis of Individual Phenolic Compounds in Water-Suspended Particles and Surface Water during Betula and Pinus Pollen-Spreading Periods

Individual phenolic compounds present in suspended particles collected during the *Betula* and *Pinus* pollen-spreading periods (SPB and SPP as bound PC) and surface water samples (SPBW, SPPW and PW) were analyzed by high-performance thin-layer chromatography (HPTLC) and high-performance liquid chromatography methods (HPLC/DAD). Standard substances of phenolic compounds (gallic, vanillic, *trans*-ferulic, p-coumaric, p-hydroxybenzoic, 3,4-dihydroxybenzoic (3,4-DHB), sinapic, syringic, and chlorogenic acids and flavonoids rutin, quercetin) were determined individually and as a mixture (1 mg/mL) by HPTLC and HPLC/DAD in parallel (Appendix A) [21] (Appendix A). The results were compared with each other.

HPTLC results showed that 3,4-DHB-acid and chlorogenic acid were the most common phenolic acids in surface waters (PWs, SPBWs, SPPWs) during this period (Figure 3 and Appendix A). Notably, a mixture of phenolic acids with Rf = 0.8 was observed. However, the data from the phenolic acid standards revealed that vanillic, *trans*-ferulic, p-coumaric, p-hydroxybenzoic, sinapic, and syringic acids had similar retardation factors, making it difficult to separate them into individual phenolic acids in the extracts. Additionally, the SPB collected during the *Betula* spreading period found gallic acid (FQK, BSC, CLC) and flavonoid quercetin (BSC and NLS). The SPP extracts contained chlorogenic acid and showed phenolic acid mixture lines (Appendix A).

In order to determine the composition of the phenolic acid mixture, water-suspended particles and surface water samples were analyzed for high-performance liquid chromatography with a diode array detector (HPLC/DAD). The results indicated that vanillic and *trans*-ferulic acids were consistently present in all samples, irrespective of the water body (Table 3).

The findings obtained using HPLC/DAD were mainly in line with the results of the HPTLS analysis. Specifically, during the *Betula* spreading period, quercetin was detected in SPB and SPBW samples. Furthermore, SPBW samples exposed traces of chlorogenic acid and hydroxybenzoic acids. Gallic acid was present in SPP and SPBW samples upon *Pinus* pollen dispersal. In addition, the presence of hydroxybenzoic acids and p-coumaric acid was detected in small amounts in the SPP extracts (Table 3).

### 2.5. Release of Phenolic Compounds from Betula and Pinus Pollen into the Water: A Laboratory Experiment

A laboratory experiment was carried out to complement the results of releasing phenolic compounds from pollen into the water. The objective of the experiment was to determine the content of phenolic compounds released by *Betula* and *Pinus* pollen and their rate of release into water. The pollen was exposed to distilled water for 24, 48, and 72 h. Phenolic compounds were analyzed both in water-exposed pollen (in the cases of *Betula*, pollen was marked as DSPB1-3, and *Pinus* pollen was marked as DSPP1-3) and in distilled water to which *Betula* and *Pinus* pollen (marked as DSPBW1-3 and DSPPW1-3, respectively) were exposed. Similarly, pollen from *Betula* and *Pinus* plants collected in situ was analyzed as a control. DSPB and DSPP extracts were analyzed for phenolic compounds that were accumulated in plant cells as free phenolic compounds (free PC) and naturally released into the water.

#### 2.5.1. Total Phenolic Content and Antioxidant Activity

The study showed that the *Betula* pollen exposure on the water surface was relatively short. The pollen deposition rate was 4 mm/min. Approximately 90% of the 500 mg of pollen settled to the bottom within 20–25 min in a volume of 8.5–9.0 cm width and 13.5 cm height.

The total phenolic content (TPC) in *Betula* and *Pinus* pollen collected in situ (control) was 3.36 mg/mL and 0.52 mg/mL, respectively. This indicates that *Betula* pollen contains approximately five times more phenolic compounds than *Pinus* pollen (Figure 4). After 24 h of exposure of *Betula* pollen (DSPB1) to distilled water, the TPC was found to be 1.96 mg/mL. This indicated a 26% decrease in phenolic compounds compared to the control (Figure 4). After 48 h, the TPC in the DSPB2 pollen had further decreased by 14% compared to DSPB1. Subsequently, after 72 h, there was a substantial difference in the total phenolic content between DSPB2 and DSPB3 extracts, with a 34% reduction observed (1.47 mg/mL and 0.72 mg/mL, respectively) (Figure 4). Significant differences (at *p* < 0.05, Appendix A) were found between the control and DSPB3: the TPC of *Betula* pollen exposed to distilled water for 72 h decreased by 65% compared to the control (Figure 4, column d–a). The TPC was determined at 0.09 mg/mL in the distilled water, in which *Betula* pollen was dispersed for 24 h (DSPBW1). After 48 h (DSPBW2), there was a 29% increase (0.17 mg/mL), and the TPC in the water after 72 h (DSPBW3) remained similar to that after 48 h (Figure 4). No significant differences were found in the water to which the *Betula* pollen was exposed.

*Pinus* pollen was also kept in distilled water under similar conditions as *Betula* pollen. The *Pinus* pollen formed small agglomerates and remained on the water’s surface. During periodic water fluctuations, the pollen was mixed and floated to the surface, with only a small amount settling to the bottom. *Pinus* pollen dispersed in distilled water for 24 h (DSPP1) lost about 22% of its phenolic compounds: from 0.52 mg/mL in the control to 0.32 mg/mL in the DSPP1. After 48 h and 72 h, the pollen (DSPP2 and DSPP3) significantly (at *p* < 0.05) lost ~30% of its phenolic compounds (up to 0.13 mg/mL and 0.14, respectively) compared to the control.

In the water to which *Pinus* pollen was exposed (DSPPW), the phenolic content showed some variations. Within 24 h, the *Pinus* pollen released about 0.1 mg/mL of phenolic compounds into the distilled water (DSPPW1). A significant difference (*p* < 0.05) in TPC was observed between DSPPW2 and DSPPW3 extracts: after 72 h, TPC in DSPPW3 decreased by 7% as compared to DSPPW2 (Figure 4, Appendix A).

Based on the results of DSPPW and DSPBW extracts, it was found that *Betula* and *Pinus* release similar phenolic compounds into the water over 24 h. However, *Pinus* pollen does not release more phenolic compounds when left in water for a longer duration. In contrast, *Betula* pollen releases about 0.07 mg of phenolic compounds after 48 h. The decrease in TPC in *Pinus* pollen after 72 h is probably due to the activity of microorganisms, which were relatively more active in *Pinus* pollen than in *Betula*.

Antioxidant activity analysis was carried out to study the effect of *Betula* and *Pinus* pollen dispersion in distilled water for different periods and the compounds released into the water. During the exposure of *Betula* pollen to distilled water, a decrease in the bioactivity of the extracts was observed; however, no significant differences were found according to the DPPH• scavenging activity (Figure 5). The antioxidant activity decreased from 58% in the control to 34% in the DSPB3 extract. The antioxidant activity of the DSPB1 extract was about 22% higher than that of the DSPB2 extracts and similar to the DSPB3 extract. The study shows that the *Betula* pollen releases bioactive compounds into the water. The antioxidant activity values ranged from 64 to 71%, with no significant differences between them (Figure 5, Appendix A).

Based on the ABTS radical-binding assay, the *Betula* pollen showed the highest antioxidant activity when exposed to distilled water for 24 h. This activity was almost equal to the control. However, the antioxidant activity of the pollen decreased when it was exposed to distilled water for extended periods and was below the method’s limit of detection (20–80%). The water to which the pollen was exposed for 72 h had the most significant antioxidant activity, which was significantly (*p* < 0.05) higher than the water to which the pollen was exposed for 48 h (Figure 5, Appendix A).

According to a study on *Pinus* pollen, the antioxidant activity, measured by DPPH• and ABTS• radical-binding intensity, was below the method’s detection limit. This was observed regardless of the exposure time of *Pinus* pollen in distilled water and the content of phenolic compounds released into the water.

#### 2.5.2. Individual Phenolic Compounds in *Betula* and *Pinus* Pollen Exposed to Distilled Water

HPTLC results show that the diversity of phenolic compounds released into distilled water (Figure 6) is similar to that from natural and artificial water sources (Figure 3 and Appendix A). Notably, the antioxidant activity of chlorogenic acid in the control, DSPB, DSPBW, and DSPPW extracts was observed, along with the bioactivity of the 3,4-DHB acid and phenolic acid mix. However, the content of bioactive compounds was low in the DSPP extracts (Appendix A).

The pollen and distilled water samples were also analyzed for flavonoids and their bioactivity. In the *Betula* pollen cases, rutin and its potential biological activity (Rf = 0.25, track 1–2) were detected in the control (Figure 6) and in the pollen exposed for 24 h in distilled water (DSPB1, track 3). Traces of rutin were only detected in DSPB2-3 and DSPBW1-3 (tracks 4–8), alongside a line of quercetin (Rf = 0.8, tracks 4–8). After 24 to 72 h of water exposure, rutin was no longer present in the *Pinus* pollen. Nevertheless, signals for rutin and quercetin were detected in the water in which *Pinus* pollen was placed (DSPPW1-3), with Rf values of 0.2 and 0.8, respectively. Additionally, other compounds were detected in the DSPPW extracts, which were not found in the control or DSPP1-3 extracts.

According to the HPLC/DAD data, the ratio of individual phenolic compounds changes when *Betula* pollen is exposed to distilled water (Figure 7). Both the pollen (DSPB) and water to which the pollen was exposed (DSPBW) contained chlorogenic acid. In the control and the pollen exposed to distilled water (DSPB), the average concentration of chlorogenic acid was 65.2 µg/mL (Figure 7a).

In the water to which the pollen was exposed (DSPBW), chlorogenic acid content increased from 0.39 µg/mL (DSPBW1) to 1.94 µg/mL (DSPBW3) (Figure 7b). The ferulic acid concentration in pollen samples after 24 h of exposure (DSPB1) was 103 µg/mL, which was close to the control (119 µg/mL). However, after 48 and 72 h of exposure, no ferulic acid was detected in the extracts of pollen samples (Figure 7a). In the DSPBW samples, ferulic acid content varied around 0.77 µg/mL concentration. The relative ferulic acid content in the water to which the pollen was exposed for an appropriate time was found to be between 24% and 40% of the total compounds identified according to method’s limit of detection (LOD) (Figure 8).

Vanillic acid was detected in the extracts, and its relative content varied from 5% to 2% in DSPB extracts (Figure 8). In DSPBW extracts, traces of vanillic acid (~1%) were also detected. Gallic acid traces were identified in the extracts of DSPB1 and DSPBW1-3.

Data of the HPLC analysis of flavonoids in pollen samples correlated with HPTLC results. The *Betula* pollen exposed to distilled water for 24 h contained rutin. Its concentration was 130 µg/mL, and it accounted for 10% of the total compounds found in the DSPB1 extract. In the water to which the pollen was exposed (DSPBW1), the concentration of rutin was found to be 0.19 µg/mL (Figure 7b). However, after 48 and 72 h of pollen exposure, rutin was not detected in DSPB and DSPBW. Traces of quercetin were detected in *Betula* pollen samples after 24 h of exposure to distilled water. DSPB1 and DSPBW1 extracts contained up to 2% quercetin. After 48 and 72 h, no quercetin was detected in DSPB2-3 extracts. In the water to which the *Betula* pollen was exposed (DSPBW2-3), the relative content of quercetin was found to be ~3%. A trend was revealed that in DSPB 1-3 extracts, unidentified phenolic compounds or derivatives increased from 20% to 50%.

In the case of *Pinus*, the analysis revealed low levels of individual phenolic compounds in the samples. After 24 h and 48 h exposure of pollen to distilled water, only traces of phenolic compounds were detected. After 72 h, traces of vanillic acid were identified in the DSPP3 extract. The distilled water to which the *Pinus* pollen was exposed for 72 h (DSPPW3) contained traces of vanillic acid, gallic acid, hydroxybenzoic acids, and rutin. The composition of phenolic compounds was similar to that found in water-suspended particles, collected during the *Pinus* pollen-spreading period. Additionally, all extracts contained unidentified phenolic compounds or derivatives.

## 3. Discussion

This study presents novel findings on the enrichment of water bodies with bioactive phenolic compounds from *Betula* and *Pinus* pollen. The findings build on previous research by providing details on the quantity and variety of phenolic compounds present in pollen collected in situ [21]. In nature, some pollen from anemophilous plants is deposited in water bodies and becomes part of the food chain [4,5,31]. Satellite imagery (Figure 1) has shown that *Pinus* pollen could be dispersed on the surface of coastal areas of the Baltic Sea. However, a high amount of pollen at the water surface can make it difficult to accurately estimate chlorophyll concentrations using remote-sensing image algorithms, as pollen increases water reflectance [32]. In this research, water sampling sites were chosen near the shore of which *Betula* and/or *Pinus* plants were growing. However, part of the pollen could enter the water from sufficiently distant areas, as shown by the presence of *Pinus* pollen in all water bodies before the pollen-spreading season in Lithuania. The study shows that only *Pinus* pollen was visible on the Baltic Sea coast near Lithuania during the pollination period (Figure 1a,b). The pollen deposited in the water was quite densely concentrated on the shore (Appendix A) and influenced low levels of other impurities in the water-suspended particles (SPP) (Table 1). During the *Betula* pollen spreading, no pollen was visible on the water body’s surface, resulting in water-suspended particles (SPB) of *Betula* pollen ranging from 3% to 50% in the samples. In all cases (both *Betula* and *Pinus* pollen-spreading period), the weather was relatively calm during the last 10 days before sampling: the average wind speed at a height of 10 m at the nearest meteorological stations varied from 1.7 to 2.9 m/s (Appendix A). In small Lithuanian lakes, the speed of wind-induced surface water currents is usually less than 1% of the wind speed measured at meteorological stations [27], and horizontal water movements in lakes and ponds during the study should not have exceeded 0.03 m/s. In the Baltic Sea and the Curonian Lagoon, the water dynamics at such wind speeds are much more intense and complex: not only surface currents but also compensatory currents form here [29]. In lakes and ponds, only the transport of the surface water layer is possible (the counter currents that may appear in deeper layers are not formed). During the *Betula* pollen season, W-NW winds prevailed. This may have resulted in higher pollen counts on the eastern and south-eastern shores of small water bodies (samples were taken on the windward shore of lakes Kalniškiai (FQK), Geluva (NLG), and Šermukšnynas (NLS). During the *Pinus* pollen season, winds from E-PW prevailed, so currents could carry more pollen towards the western and north-western coasts (where samples were taken in lakes Juodlė and Damba). The results of the sample analysis show that the amount of pollen among the water-suspended particles in the samples is strongly influenced by wind direction during sampling, with minimal dependence on meteorological conditions over the past ten days. Determining the proportion of pollen in water samples that is carried by flowing water remains challenging.

The lake’s environment may influence the release of chemicals from pollen in the water [4,5]. Pollen rich in biogenic nutrients [33], when introduced into the aquatic ecosystem, replenishes it with a significant amount of carbon, phosphorus, nitrogen [4,5,34], and fatty acids [35], and it has a positive influence on algal growth and zooplankton abundance [3]. Water bodies contain phenolic compounds that enter the water as natural decomposition products of dead plants and animals [36]. Consequently, the surface water of all studied water bodies (PW, SPBW, SPPW) contained between 0.1 mg/mL and 1.70 mg/mL dissolved phenolic compounds during the study period. This variation in the data could be due to other sources of phenolic compounds (Appendix A) in the samples [17,37].

The content of phenolic compounds in the SPP water-suspended particle samples remained similar, irrespective of whether they were present in free (free PC) or bound (bound PC) form. In some cases, the SPB particles contained fewer phenolic compounds as free PC than bound PC. However, an in situ study of *Betula* pollen [21] showed that free PC is 20% higher than bound PC. The low amount of free PC in SPB extracts is likely because pollen releases substances from the cell cytoplasm into water [9]. Bound PC and other nutrients are diffused into the water and consumed by microorganisms during pollen degradation [5]. Some of these compounds end up in the bottom sediments with pollen [38]. According to [39], *Betula* pollen needs at least three hours to rupture in natural water. Another study [5] indicates that the decomposition of pollen in water bodies is primarily due to released chemicals. Our laboratory experiment showed that *Betula* and *Pinus* pollen release a similar total content of phenolic compounds into the water within 24 h. After 48 h, the *Betula* pollen released more phenolic compounds than *Pinus*. The differences were hardly noticeable in the case of *Pinus*. Phenolic compounds from the *Betula* pollen probably enter the water from the cytoplasm [9] through pores in the pollen exine [40]. Phenolic compounds from the *Pinus* pollen are likely to enter the water from the pollen surface, because the exine pores are tiny, and covalent bonds do not bind phenolic compounds [41]. Upon entering the aquatic environment, the *Pinus* pollen is quickly colonized by aquatic microorganisms and decomposes within a few days [3,5]. Our results showed that some of the *Pinus* pollen was already in the decomposition stages when we collected samples from hypereutrophic water bodies’ surfaces (CLC, DFD), and the total phenolic compounds in these water body extracts were lower than in others. This could be due to the higher activity of microorganisms, which assimilate some of the chemicals released during the first stage of pollen decomposition [5]. Released compounds may serve as a food source for local invertebrates [42].

The antioxidant activity of the compounds is routinely evaluated by the ABTS (2,2’-azino-bis(3-ethylbenzothiazoline-6-sulfonic acid) radical cation) and DPPH (2,2-diphenyl-1-picrylhydrazyl) radical-scavenging methods. Their reactivity can be influenced by the steric effects [43]. The study shows that regardless of the water body, water-suspended particles and surface water bioactivity were low (Appendix A). The presence of water-suspended particles and surface water matrices can likely affect the antioxidant activity of compounds. Particles can serve as surfaces for radical reactions and alter the accessibility of radicals to antioxidants [44]. The chemical structure of each compound determines its specific mode of action in neutralizing ROS [43]. Therefore, the chemical composition of particles suspended in water and dissolved organic matter in the aquatic ecosystem can influence radical scavenging [45]. Bioactive compounds are accumulated by marine-derived fungi, cyanobacteria, microalgae, seagrasses, and sponges [17,37]. Studies on bioactive compounds show that plant essential oils are not toxic to relevant shellfish species [46] and have no significant effect on the fish feeding quality; however, they improve the organism’s protective effect against damage to the cells [47]. Our laboratory experiment has shown that the *Betula* pollen releases stable or modified forms of compounds with high bioactivity into the water. Phenolic compounds have aromatic rings with hydroxyl groups and other substituents, which are modified by oxidative enzyme action [13]. Different modifications, such as hydroxylation, glycosylation, methylation, and acylation, play an essential role in generating the diversity of flavonoids and contribute to their improved solubility and stability and their bioactive properties [48]. Significant release from the *Betula* pollen into distilled water consisted of chlorogenic acid, which is also found in natural water bodies. Chlorogenic acid is soluble and relatively stable in aqueous solutions [49]. It retains high antioxidant activity and generally exhibits higher antioxidant activity compared to gallic acid [14]. However, as chlorogenic acid undergoes modifications, its antioxidant activity decreases [48]. The bioactivity of chlorogenic acid is based on structural differences between the compounds and the presence of additional hydroxyl groups [14,49]. The two primary phenolic acids in the water-suspended particles and surface water of the studied water bodies were trans-ferulic and vanillic acids. Ferulic acid is an antioxidant commonly found in plant cell walls, mainly conjugated with mono- and oligosaccharides, polyamines, lipids, and polysaccharides, and it seldom occurs in a free state in plants [50,51]. Ferulic acid plays a vital role in providing rigidity to the cell wall and forming other important organic compounds like coniferyl alcohol, vanillin, sinapic, diferulic acid, and curcumin [52]. Vanillic acid can be found in a variety of plants, including aquatic ones [17]. Our laboratory experiment showed that ferulic acid was detected in pollen, collected in situ (control) in pollen (DSPB1) and the water (DSPBW1) to which the *Betula* pollen was exposed for 24 h. Traces of vanillic acid were found in pollen and distilled water (2–4%). Other studies have shown that ferulic acid can be converted to vanillic acid [53,54]. Rutin was not detected in the water samples taken from the investigated natural water bodies. It was released only in distilled water, to which the *Betula* pollen was exposed for 24 h. Rutin is a precursor of quercetin [55,56], and the content of quercetin in distilled water slightly increased. It is possible that rutin underwent some other modification [57] such as hydroxylation, which is common in natural flavonoids [48]. This may have resulted in an increase in unidentified compounds in the *Betula* pollen (Figure 7b).

During the *Pinus* pollen-spreading period, the concentration of phenolic acids in water-suspended particles, surface water, and pollen was below the limit of quantification. The water-suspended particles and surface water contained only small traces of gallic acid, hydroxybenzoic acid, and p-coumaric acid. Vanillic and p-coumaric acids have low antioxidant activity [14]. In our study, analogous results were obtained with the HPTLC method for the derivatization of these phenolic acids with the DPPH• (Appendix A). In addition to phenolic acid mixtures, antagonistic effects could be observed for protocatechuic acid with gallic acids and protocatechuic acid with vanillic acids [14]. Thin-layer chromatography is a quick and simple method for obtaining important information about the qualitative composition of tested samples when the concentration in samples is low.

It has been shown [58] that even when the *Pinus* pollen wall is destroyed, there is no substantial increase in the content of phenolic compounds in the extracts. *Pinus* produces large quantities of pollen [3,58], which is dispersed in the environment and deposited in natural waters [32]. In nature, water bodies typically have different trophic levels [27]. Our study shows (Table 2) that during the period of pollen spreading, the total content of phenolic compounds in water-suspended particles and in surface water varies. The analysis of samples collected during *Betula* and *Pinus* pollen spread shows ambiguous effects on water bodies of different trophicity: the total phenolic compounds in dystrophic water bodies differ from those in eutrophic, hypereutrophic, and mesotrophic water bodies (Table 4).

In dystrophic lakes, low pH may suppress bacterial metabolism, and primary production is reduced due to light attenuation by refractory dissolved organic carbon [59]. The trend from our study revealed that a higher phenolic compound content was obtained in the water bodies, indicating a dystrophic state. In some cases [60], the uptake of nutrients released by pollen can slow down and gradually contribute to the dystrophic transformation of the lake ecosystem. Those researchers state that higher levels of tannins inhibit the nitrification of ammonium ions in the water, while at the same time suppressing the intense growth of phytoplankton and higher aquatic plants. Specifically, the organic matter formed within the lake is the main factor of natural eutrophication [27].

The low levels of phenolic compounds released into the natural water might have limited their possible impact on the change in the trophic state of water bodies. It is challenging to evaluate the characteristics of pollen transport and distribution in the natural waters. The results of other researchers’ studies [37,61] show that the pollen (as well as other bottom sediments) deposited in different places of water bodies is varies in size and weight. In Lithuanian lakes, *Pinus* pollen is mostly found in the sediments of the shallow water zone, whereas *Betula* pollen is found in the deep part of the bottom [62,63]. Pollen is one of the smallest and lightest particles suspended in water, so most of it tends to settle in those parts of lakes where sediments with the lowest hydraulic weight accumulate. The pollen particles likely travel the greatest distances in the water under the influence of surface currents and waves. Such an assumption can be made based on the results of sedimentation process research in the lakes of Lithuania and neighboring regions: in many lakes (especially in smaller ones), the mechanism of transport of small particles is linked specifically to the dynamics of the surface water layer [38,64,65]. It has also been observed that pollen that has sunk in shallow lakes can be repeatedly lifted from the bottom and transported to another location.

## 4. Materials and Methods

### 4.1. Analysis of Betula and Pinus Pollen Data and Information Gathered by Satellites

Airborne pollen data of *Betula* and *Pinus* pollen were obtained from Vilnius University aerobiological monitoring stations in Šiauliai and Klaipėda. The average pollen concentration was calculated by estimating the intensity of pollen spreading five days before water sample collection. According to the water sampling time in the Klaipėda region, *Betula* pollen data analysis covered dates from 25 to 29 April 2022. The study of *Pinus* pollen data included the period from 4 to 8 June 2022. The daily average of the *Betula* pollen concentration was 205 pollen/m^3^ between 25 April 2022 and 29 April 2022, and the *Pinus* pollen concentration was 278 pollen/m^3^ between 4 June 2022 and 8 June 2022.

In Šiauliai, the *Betula* pollen-spreading period was analyzed using airborne pollen data and ranged from 30 April to 10 May 2022. The *Pinus* pollen-spreading period ranged from 31 May to 8 June 2022. The average *Betula* pollen concentration was 653 pollen/m^3^ (30 April 2022–4 May 2022), 921 pollen/m^3^ (2 May 2022–6 May 2022), 1561 pollen/m^3^ (4 May 2022–8 May 2022), and 1331 pollen/m^3^ (6 May 2022–10 May 2022). The average *Pinus* pollen concentration was 324 pollen/m^3^ (31 May 2022–4 June 2022), 313 pollen/m^3^ (2 June 2022–6 June 2022), and 243 pollen/m^3^ (4 June 2022–8 June 2022).

Remote-sensing data help in the observation of geobiophysical environmental variables, including pollen in water [4]. We used the optical medium-resolution Sentinel-2 MultiSpectral Imager data from the Copernicus program and high-resolution Planet Lab Inc. data from PlanetScope Super Dove satellites. Sentinel-2 satellites have a spatial resolution of 10 m, and PlanetScope satellite data are of 3.7 m spatial resolution. Sentinel-2 data were obtained through Google Earth Engine python API [66], and Planet data were downloaded through the Education and Research program of Planet Labs PBC [67]. The satellites of both missions have spectral bands in visible and near-infrared parts of the electromagnetic spectrum, and the Sentinel-2 bands also extend to short-wave infrared. We used products containing atmospherically corrected surface reflectance. Aggregations of pollen are visible in the sea; thus, we investigated the images visually and found both satellite images on 9 June 2022, the same day as the in situ collection of pollen near Giruliai was carried out. In addition, the PlanetScope satellite passed the location earlier in the day and captured cloudless views of the area, whereas the Sentinel-2 image was taken an hour later, and some pollen slicks were covered by clouds. The surface reflectance of pollen slicks was extracted from images, and their spectra were plotted in a log-transformed manner.

### 4.2. Natural and Artificial Water Bodies’ Characteristics

Water-suspended particles and surface water samples were taken from 10 different locations. Two sampling sites were in the Baltic Sea and the Curonian Lagoon. The other 6 sampling sites were located in 6 small lakes in the north-western part of Lithuania (one sampling site was selected for each lake). The location of the research places is shown in Figure 9.

The main characteristics of the studied water bodies are described in Table 5. In order to describe the possibilities of bringing pollen into water bodies and the characteristics of their distribution in the water mass, the prevailing meteorological conditions at the sampling sites were analyzed. Meteorological conditions are described using the data of the nearest meteorological station for the last 10 days (prior to the moment of sampling) (Appendix A).

The trophic status of water bodies was assessed according to data provided by other authors and external water bodies’ signs. Detailed studies allowing for the determination of the trophic level with sufficient accuracy have been carried out several times only in the Baltic Sea and the Curonian Lagoon [68,69]. Among the lakes and ponds analyzed in this study, the trophic status of the Lake Bijotė was accurately assessed based on biological and chemical indicators [70]. The trophic status of other inland water bodies was assessed during the inspection according to typical trophic indicators: overgrowth of coastal zones and open-water areas with aquatic plants, thickness of the silt sediment layer, optical properties of the water, etc. The descriptions of the research carried out in these objects in previous years [71,72] were additionally used for the assessment of their trophic status.

### 4.3. Sampling of Water-Suspended Particles and Surface Water during Betula and Pinus Pollen-Spreading Periods

Five types of samples from different water bodies were prepared:SPB—water-suspended particles collected from the surface water of different water bodies during the *Betula* pollen-spreading period.SPBW—surface water samples filtered and collected in parallel with SPB sampling. SPB and SPDW samples were collected from 29 April to 10 May 2022.SPP—water-suspended particles collected from the surface water of different water bodies during the *Pinus* pollen-spreading period.SPPW—surface water samples filtered and collected in parallel with SPP sampling. SPP and SPPW samples were collected from 4 to 8 June 2022.PW—clear surface water samples were collected before *Betula* and *Pinus* pollen spreading. The surface water samples were collected from 20 to 23 March 2022.

An automatic water-suspended particle collection system was installed. A 12 V pump was placed in a buoy and submerged into the water at a depth of 2–3 cm (Figure 10). An external energy source, a battery, was used to pump surface water into the sieve system. For sample collection during the *Betula* pollen-spreading period, the water was filtered through sieves with 80 µm and 40 µm eyelets (Figure 10). The suspension was collected on the sieve, where eyelets were 20 µm in size. Surface water was pumped at a distance of about 10–15 m from the coast of water bodies.

During the *Pinus* pollen-spreading period, the surface water was filtered through sieves with 125 µm and 80 µm eyelets. The suspension was collected on the sieve, where eyelets were 40 µm in size. Surface water was pumped from the coast of water bodies.

The water-suspended particles and surface water filtrate were poured into 1 L containers, with at least two replicates, acidified with 2 M sulphuric acid to pH 2, and transported to the laboratory. The water-suspended particles were rinsed with distilled water and dried in a thermostat at 40 °C until dry mass. Surface water filtrate was additionally filtered through a 0.8 µm membrane filter and concentrated using solid-phase extraction. All samples were stored at +4 °C until analysis.

At least eight samples of suspended particles and surface water were collected from water bodies during the study period. A total of 72 samples were collected: 32 samples of water-suspended particles and 40 samples of surface water.

### 4.4. Water-Suspended Particle and Surface Water Extract Preparation

The extracts of water-suspended particles, collected during the *Betula* and *Pinus* pollen-spreading periods, were prepared for phenolic compounds bound to the plant’s cell wall (bound PC) and present as free phenolic compounds (free PC). Each extract was measured using the method adapted to current study and previously described in available reference [21].

Bound-to-the-cell-wall phenolic compound (bound PC) extracts were prepared using 100 mg dry weight of water-suspended particles collected from different water bodies during *Betula* and *Pinus* pollen spreading was poured into 2.5 mL of 0.1 M sodium hydroxide. Then, it was shaken for one hour in an orbital shaking device (Heidolph Incubator 1000, Schwabach, Germany) at 40 °C, cooled to room temperature, and acidified with 2 M of hydrochloric acid to pH 5–6. Up to 5 mL of methanol was added and shaken in an ultrasonic bath for 30 min. Then, the sample was cooled, left to clarify, filtered through a 0.22 µm membrane filter (Chromafil ^®^Xtra PTFE-20/13, Düren, Germany), and stored until analysis at +4 °C.

Free phenolic compound extracts were prepared using 100 mg dry weight of water-suspended particles collected from different water bodies during *Betula* and *Pinus* pollen spreading and poured with 70% methanol at a ratio of 1:10, incubating in an orbital shaking device (Heidolph Vibramax 100, Schwabach, Germany) with a constant duration of shaking. The samples were then centrifuged (Eppendorf Centrifuge 5702, Hamburg, Germany), clarified, filtered, and stored in a refrigerator (Snaigė RF59FB-P500270, Alytus, Lithuania) at +4 °C until analysis.

Surface water extracts were prepared using the solid-phase extraction method. One liter of filtered surface water was concentrated using a solid-phase extraction disk with a diameter of 47 mm and sorbent active group C18, containing 500 mg of carbon (Empore™, Merck KGaA, Darmstadt, Germany). The ENVI-Disk™ Holder (Merck KGaA, Darmstadt, Germany) appropriate for the diameter of the disk was used for the samples’ extraction. Solid-phase extraction was performed according to the sorbent manufacturer’s recommendations. The sorbent was washed twice with 5 mL of pure methanol and dried for 3 min after each wash. For the sorbent conditioning, 5 mL of pure methanol and 5 mL of 10% methanol was added without allowing the sorbent to dry. When a layer of 2–3 mL of methanol solution remained on the sorbent, a water sample was extracted. A concentration of 70% methanol was used to desorb the analytes in two 2.5 mL replicates. The final volume of the extract was 5 mL, which was stored at 4 °C until analysis.

### 4.5. Water-Suspended Particles’ Composition Microscopy

One drop of water with suspended particles was dropped onto a 90 mm × 20 mm slide. The slide was divided into six fields. The fields were analyzed using a Leica ×400 magnification light microscope (Danaher, Wetzlar, Germany). Pollen and impurities were counted. The average pollen and impurity content of the sample was calculated as a percentage. Pollen was recognized according to distinct morphological characteristics and at the taxonomic level of the plant family or genus.

### 4.6. Water-Suspended Particle and Surface Water Extract Chemical Analysis

The total phenolic content and antioxidant activity using DPPH• (2,2-Diphenyl-1-picrylhydrazyl) and ABTS• (2,2-azino-bis (3-ethylbenzthiazoline-6-sulfonate) radical-scavenging were determined using the spectrometric method. Individual phenolic compounds were identified using high-performance thin-layer and high-performance liquid chromatography (HPLC-DAD) methods. Extracts were analyzed using the adapted versions of methods and the same equipment as described in a previous study [21]. The HPLC-DAD analysis was performed to complement the HPTLC results and determine phenolic compounds’ distribution in water-suspended particles and water. Spectra of reference materials for phenolic compounds at a concentration of 1 mg/mL were used to identify phenolic compounds in the extracts [21]. The peak intensities of the phenolic compounds in the UV range of 250, 260, 280, and 310 nm were evaluated. Based on the peak intensities, the wavelengths for the analysis of phenolic compounds were selected: 260 nm—trans-ferulic, vanillic, gallic, p-hydroxybenzoic, 3,4-dihydroxybenzoic, p-coumaric acids, rutin, quercetin; 310 nm—syringic acid, chlorogenic acid. A calibration curve was established for each phenolic compound to quantify the concentration. The quantity intervals for trans-ferulic acid, chlorogenic acid, rutin, and quercetin quantification ranged from 0.10 µg to 1.0 µg, and the quantity intervals for vanillic acid and gallic acid ranged from 0.25 µg to 1.0 µg. The calibration data of phenolic compounds are presented in Appendix A.

### 4.7. Preparation of Laboratory Experiment of Betula and Pinus Pollen Phenolic Compounds Exposed to the Distilled Water

A laboratory experiment was carried out to analyze the release of phenolic compounds from *Betula* and *Pinus* pollen into water. The 500 mg of *Betula* and *Pinus* pollen was exposed to 1 L of distilled water for 24, 48, and 72 h. To prevent the impact of microorganisms, sulphuric acid was added to the distilled water to pH 3–4. *Betula* and *Pinus* pollen was vacuum filtered from the water on a 0.8 µm membrane filter. The filter with the pollen was placed in a thermostat and dried to dryness at 40 °C. The water to which the pollen was exposed was concentrated by the solid-phase extraction method to a volume of 5 mL, as described in Section 4.5. A total of 39 samples were prepared: 18 pollen samples exposed to water, 18 samples of distilled water after pollen exposure, and 3 samples of control (pollen collected in situ). A series of pollen extracts were prepared using the free phenolic compound extraction method as described in Section 4.5. The extracts were labeled according to the following:

Pollen collected from *Betula* and *Pinus* trees in situ was utilized as a control in the research. Part of these pollen samples was separated and dispersed in distilled water to measure the release of bioactive compounds into the water.

DSPB1, DSPB2, and DSPB3 are the labels given to the *Betula* pollen exposed to distilled water for 24, 48, and 72 h, respectively. Correspondingly, DSPBW1, DSPBW2, and DSPBW3 refer to the distilled water to which the *Betula* pollen was exposed for 24, 48, and 72 h.

The *Pinus* pollen that was exposed to distilled water for 24 h is labeled as DSPP1, 48 h DSPP2, and 72 h DSPP3. Similarly, the distilled water exposed to the *Pinus* pollen for 24 h is labeled as DSPPW1, for 48 h as DSPPW2, and for 72 h as DSPPW3.

The total and individual phenolic compounds and the antioxidant activity of all pollen and distilled water extracts were analyzed using the adapted versions of methods and the same equipment as described in earlier study [21]. For chemical analyses, 100 mg each of *Betula* and *Pinus* pollen was taken in situ (control) and exposed for the appropriate time to distilled water. Phenolic compounds in distilled water were released from 500 mg of pollen. Using spectrophotometric measurements, we calculated the final content of phenolic compounds that were released into the water from 100 mg of pollen.

### 4.8. Statistical Data Analysis

The average and standard deviation of the total phenolic content and antioxidant activity were calculated. Statistical data analysis was performed using R 3.3.0 [73] and RStudio 2023.12.0 [74] software. Several statistical tests were used to analyze whether there were any statistically significant differences between the total phenolic content in *Betula* and *Pinus* pollen exposed to distilled water for the appropriate time. In addition, differences in the antioxidant activity of the *Betula* pollen when exposed to distilled water were analyzed using the same statistical tests. Due to the small sample size, non-parametric statistics were used primarily to see whether there were statistically significant differences between variables. The Kruskal–Wallis and Friedman rank sum tests were used. As statistically significant differences were found between the samples, data were tested applying Dunn’s test with Bonferroni correction for *p*-values by using the FSA [75] package for R 0.9.5. 2023.

According to the Shapiro–Wilk normality test, total phenolic content data were checked to determine whether the distribution of the analyzed data deviated from a normal distribution. The test showed that the data generally followed a normal distribution, which resulted in the application of Welch’s *t*-test. Additionally, the application of the method was determined by the intention to compare averages from unequal variances. Welch’s *t*-test was used to analyze whether the total phenolic compounds in water bodies and their suspended particles could be related to the trophic status of the water body. Only the absolute *t*-value was shown, which does not indicate the direction of the effect. Statistically significant differences in average total phenolic content were found at *p* < 0.01 and *p* < 0.05. The data on pollen content in SPB was not normally distributed. Thus, correlation coefficients were calculated according to Spearman’s rank correlation method. The correlation between the total phenolic content and pollen content in SPB was analyzed.

## 5. Conclusions and Outlook

The study presents new evidence on bioactive phenolic compounds in open water during the flowering season of *Betula* and *Pinus*. Our research demonstrated that *Pinus* pollen remains on the water surface as suspended particles significantly longer than *Betula* pollen. Testing *Betula* and *Pinus* pollen in laboratory experiments verified that *Betula* pollen sinks at a rate of 4 mm/min, whereas *Pinus* pollen floats on the surface, possibly until particle degradation. Furthermore, the content and stability of phenolic compounds released into the water by pollen can vary over time. The period of the most substantial release occurs within 24 h after the start of exposure. In a day, roughly 26% of the phenolic compounds released into the aquatic environment are from *Betula* pollen, and 22% are from *Pinus*. The total phenolic content of *Betula* pollen continues to decrease as the pollen is exposed to water for more extended periods, whereas that of *Pinus* pollen does not change significantly. The amount of individual phenolic compounds in water samples after *Pinus* exposure is negligible. *Betula* pollen releases biologically active chlorogenic acid and rutin into the water. Chlorogenic acid exhibits durability at concentrations of 0.39–1.94 µg/mL, whereas rutin demonstrated low stability. We considered that the artificial conditions of the experiment, caused by the use of distilled water, were a limiting factor in the ex situ study. Surface water extractions revealed the presence of phenolic compounds throughout the study period, irrespective of the trophicity of the reservoir. During the study period, the total content of phenolic compounds in surface water samples varied from 0.10 mg/mL to 1.70 mg/mL. Phenolic acids, including trans-ferulic, vanillin, and 3,4-dihydroxybenzoic acid, were consistently present in surface water samples regardless of the studied water body or pollen dispersal period. Additionally, chlorogenic acid was found in the surface water of numerous water bodies. The complexity of ecosystem processes in water bodies limits our findings. Therefore, future research should focus on determining (1) the quantity of microorganisms during pollen exposure in water and (2) the ex situ emission features of phenolic compounds from pollen by collecting lake water samples during the non-vegetation season. Including this information in our results would provide a more detailed assessment of how pollen dispersed in surface water enriches water bodies with phenolic compounds.

## Figures and Tables

**Figure 1 plants-13-00099-f001:**
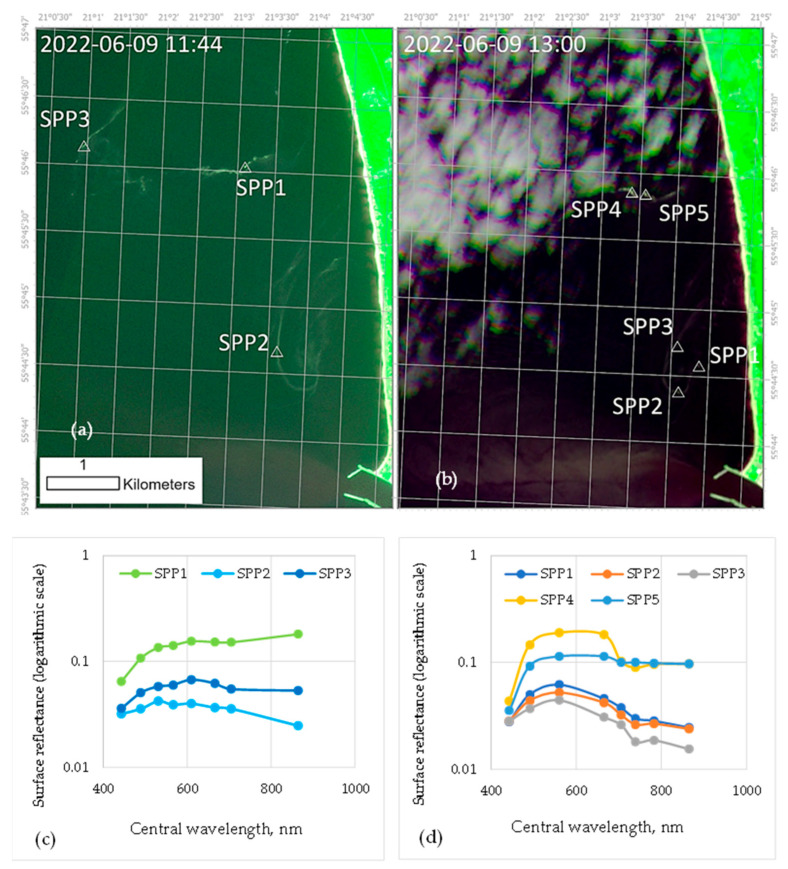
Modified RGB composites of (**a**) PlanetScope image and (**b**) Sentinel-2 image, where red color band (central wavelength 665 nm for both satellites) is assigned to red channel, near-infrared band (central wavelength 865 nm for both satellites), and blue band is assigned to blue channel (central wavelength 490 nm) of the composite. *Pinus* pollen (SPP) with numbers in the images refer to the locations of spectral signatures from (**c**) PlanetScope and (**d**) from Sentinel-2 image that show pollen aggregations.

**Figure 2 plants-13-00099-f002:**
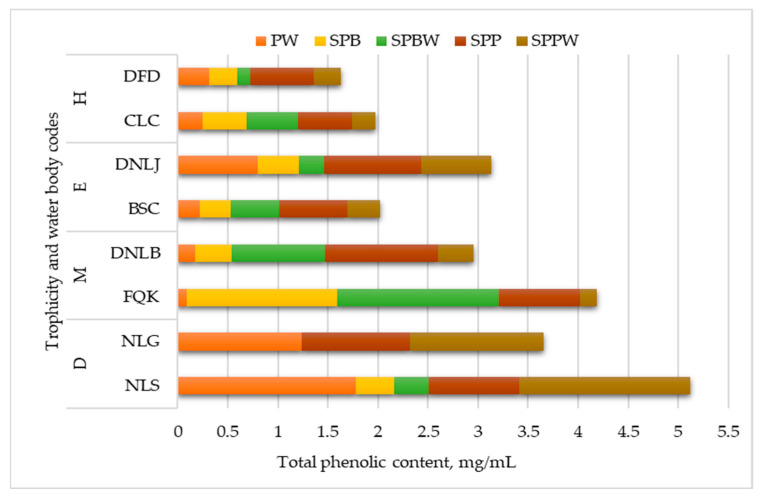
The total phenolic content in water-suspended particles and surface water during the *Betula* and *Pinus* pollen-spreading periods. Meaning of abbreviations: PW—surface water samples collected before *Betula* pollen spreading (*n* = 16); SPBW—surface water samples collected during the *Betula* pollen-spreading period (*n* = 14); SPPW—surface water samples collected during the *Pinus* pollen-spreading period (*n* = 16); SPB—water-suspended particles during *Betula* pollen-spreading period (*n* = 14); SPP—water-suspended particles during *Pinus* pollen-spreading period (*n* = 16). Water bodies’ trophicity: D—dystrophic and eutrophic with indication of dystrophy, M—mesotrophic and mesotrophic-eutrophic, E—eutrophic and eutrophic-hypereutrophic, H—hypereutrophic. The total phenolic content was expressed as rutin equivalents in mg/mL.

**Figure 3 plants-13-00099-f003:**
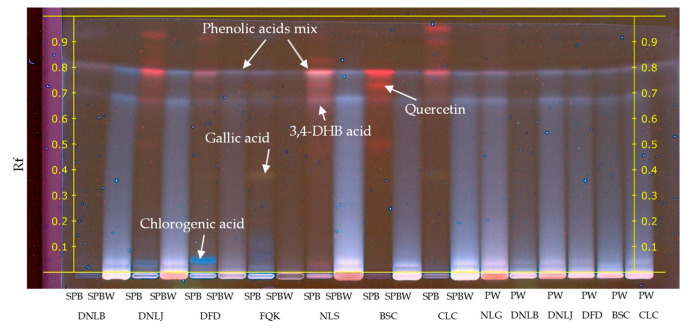
HPTLC of water-suspended particles and surface water collected from water bodies during the *Betula* pollen-spreading period. Meaning of abbreviations: SPB—water-suspended particles, collected during the *Betula* pollen-spreading period; SPBW—surface water samples, collected during the *Betula* pollen-spreading period; PW—surface water samples collected before pollen spreading. Injection volume 15.0 μL. Mobile phase: chloroform/ethyl acetate/acetone/formic acid 40/30/20/10 *v*/*v*/*v*/*v*.

**Figure 4 plants-13-00099-f004:**
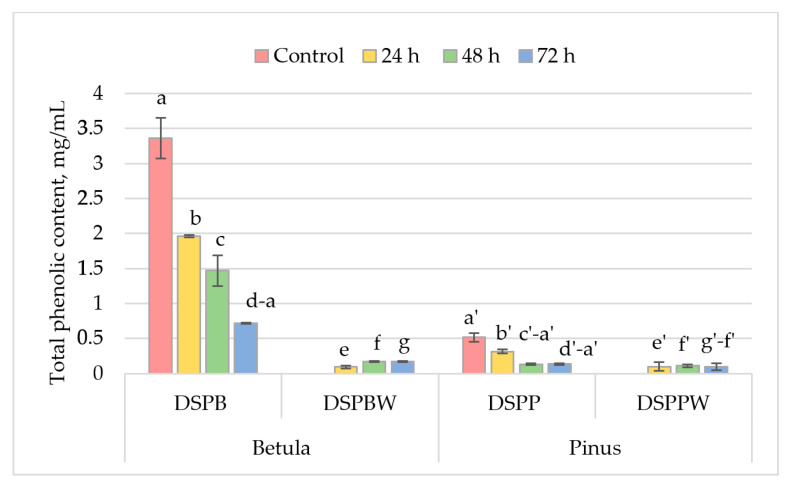
Total phenolic content in *Betula* and *Pinus* pollen exposed to distilled water for the appropriate time. The total phenolic content was expressed as rutin equivalents in mg/mL. Meaning of abbreviations: Control—total phenolic content in *Betula* or *Pinus* pollen collected in situ (*n* = 4), DSPB—phenolic compounds in *Betula* pollen exposed to distilled water for the appropriate time (*n* = 4), DSPBW—phenolic compounds in distilled water to which the *Betula* pollen was exposed for the appropriate time (*n* = 4), DSPP—phenolic compounds in *Pinus* pollen exposed to distilled water for the appropriate time (*n* = 4), DSPPW—phenolic compounds in distilled water to which the *Pinus* pollen was exposed for the appropriate time (*n* = 4). The letters on the column indicate the datasets, and double letters indicate statistically significant differences (*p* < 0.05) between variables (Appendix A).

**Figure 5 plants-13-00099-f005:**
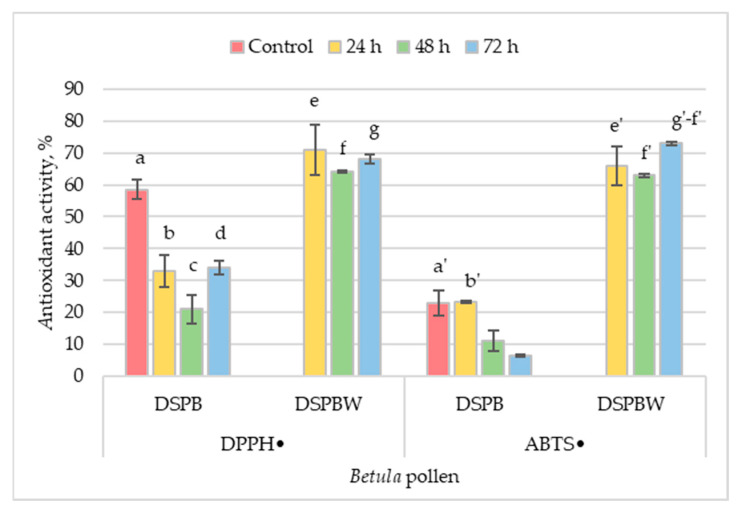
Antioxidant activity of *Betula* pollen exposed to distilled water for the appropriate time. Meaning of abbreviations: Control—total phenolic content in pollen collected in situ, DSPB—phenolic compounds in *Betula* pollen exposed to distilled water for the appropriate time, DSPBW—phenolic compounds in distilled water to which the *Betula* pollen was exposed for the appropriate time. Antioxidant activity according to 2,2-diphenyl-1-picrylhydrazyl (DPPH•) and 2,2-azino-bis(3-ethylbenzthiazoline-6-sulfonate) (ABTS•) scavenging activity. The letters on the column indicate the datasets, and double letters indicate statistically significant differences (at *p* < 0.05) between variables (Appendix A).

**Figure 6 plants-13-00099-f006:**
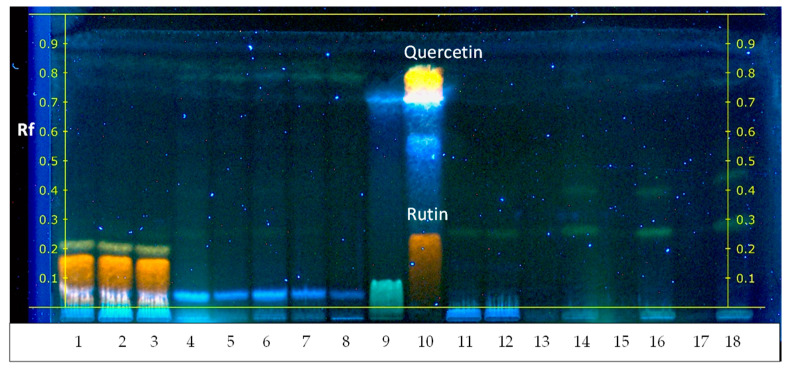
High-performance thin layer chromatography of *Betula* and *Pinus* pollen exposed to distilled water for the appropriate time after derivatization of DPPH• solution. Track indications: 1–2 control (*Betula* pollen, collected in situ)*. Betula* pollen exposed to distilled water: 3—for 24 h (DSPB1); 5—for 48 h (DSPB2); 7—for 72 h (DSPB3). Distilled water to which *Betula* pollen was exposed: 4—for 24 h (DSPBW1); 6—for 48 h (DSPBW2); 8—for 72 h (DSPBW3). 9—phenolic acid standard mix; 10—rutin and quercetin standard mix. 11–12—control (*Pinus* pollen collected in situ). *Pinus* pollen exposed to distilled water: 13—for 24 h (DSPP1); 15—for 48 h (DSPP2); 17—for 72 h (DSPP3). Distilled water to which *Pinus* pollen was exposed: 14—for 24 h (DSPPW1); 16—for 48 h (DSPPW2); 18—for 72 h (DSPPW3). The mobile phase consisted of acetone:chloroform:water (80/20/10 *v*/*v*/*v*). Injection volume: samples 10 µL, standards 2 µL.

**Figure 7 plants-13-00099-f007:**
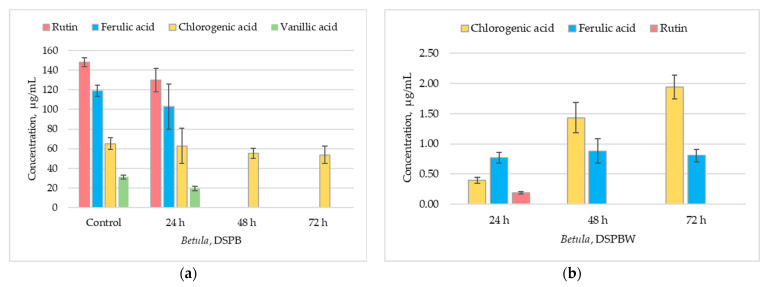
Distribution of individual phenolic compounds in *Betula* pollen and water after pollen exposure to distilled water for an appropriate time: (**a**) phenolic compounds in *Betula* pollen exposed to distilled water (DSPB), (**b**) phenolic compounds in distilled water to which the *Betula* pollen was exposed for the appropriate time (DSPBW). Mobile phase: solvent A (100% methanol) and solvent B (10% acetonitrile and 2% acetic acid in water). Flow rate—1 mL/min. Injection volume 5 µL.

**Figure 8 plants-13-00099-f008:**
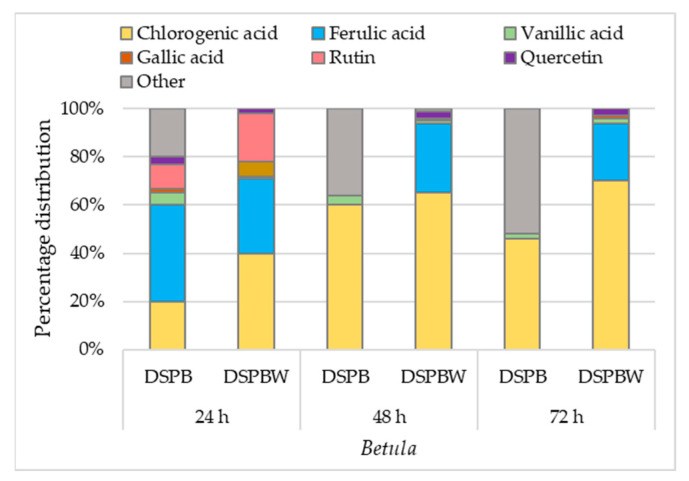
Percentage distribution of individual phenolic compounds at the limit of detection in *Betula* pollen and water after exposure of pollen to distilled water for an appropriate time.

**Figure 9 plants-13-00099-f009:**
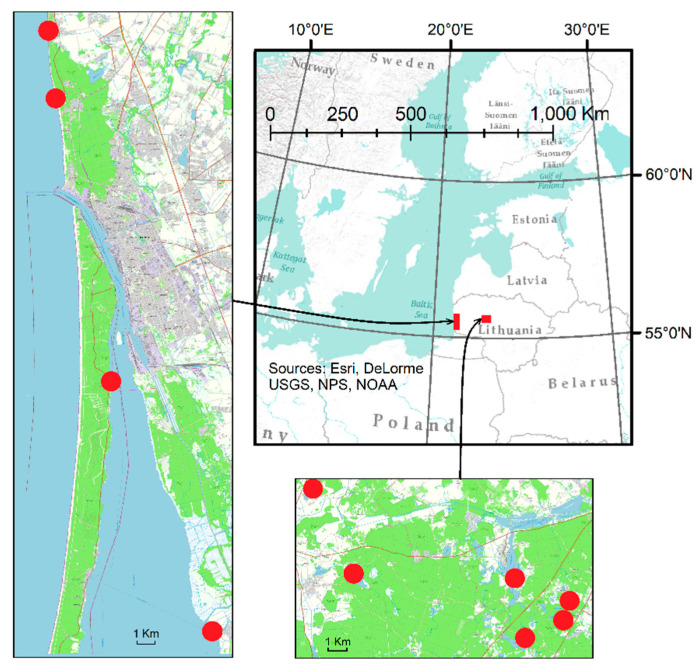
Water-suspended particles and surface water sampling location. Red squares indicate sampling location and red circles indicate sampling points in the locations.

**Figure 10 plants-13-00099-f010:**
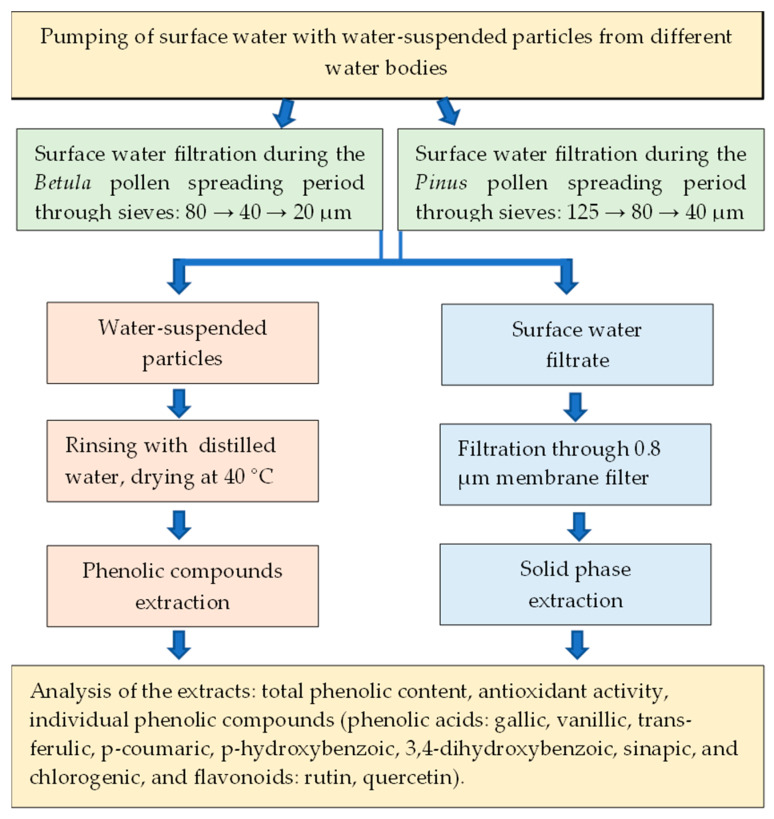
Scheme of water-suspended particle and surface water extract preparation.

**Table 1 plants-13-00099-t001:** Pollen content (percentage of pollen among all other particles) and composition (percentage of *Pinus* and *Betula* pollen in the sample) in water-suspended particles.

Trophicity of Water Bodies	Code of Water Bodies	I Stage (29 April–10 May 2022), SPB		II Stage (4 June–8 June 2022), SPP	
		Pollen Content, %	Pollen Composition	Pollen Content, %	Pollen Composition
Dystrophic	NLS	3	98% *Betula*, 2% *Pinus*	50	99.5% *Pinus*, 0.5% *Picea*
Eutrophic with indications of dystrophy	NLG	-	-	95	98% *Pinus*, 2% *Picea*
Mesotrophic-eutrophic	DNLB	10	90% *Betula*, 10% *Pinus*, *Alnus*, *Corylus*	80	100% *Pinus*
Mesotrophic	FQK	21	99% *Betula*, 1% *Pinus*, *Fraxinus*, *Alnus*	80	100% *Pinus*
Eutrophic	BSC	5	52% *Corylus*, 34% *Alnus*, 10% *Betula*, 4% *Pinus*, *Carpinus*	90	100% *Pinus*
Eutrophic-hypetrophic	DNLJ	57	99% *Betula*, 1% *Alnus*, *Corylus*	95	99.8% *Pinus*, 0.2% *Picea*
Hypertrophic	CLC	6	*83*% *Betula*, 7% *Salix*, *Fraxinus*	80	100% *Pinus*
Hypertrophic	DFD	6	91% *Betula*, 9% *Pinus*, *Alnus*	80	99.5% *Pinus*, 0.5% *Picea*

Meaning of abbreviations: NLS—natural lake Šermukšnynas; NLG—natural lake Geluva; DNLB—dammed natural lake Bijotė; FQK—flooded quarry Kalniškiai; BSC—surface water of the Baltic Sea coast; DNLJ—dammed natural lake Juodlė; CLC—surface water of the Curonian Lagoon coast; DFD—dammed fishpond Damba; SPB—water-suspended particles collected during *Betula* pollen-spreading period; SPP—water-suspended particles collected during *Pinus* pollen-spreading period.

**Table 2 plants-13-00099-t002:** The content of bound and free phenolic compounds in water-suspended particles during the *Betula* and *Pinus* pollen-spreading periods.

Water Body		SPB, mg/g (*n* = 4)		SPP, mg/g (*n* = 4)	
Trophicity	Code	Bound PC	Free PC	Bound PC	Free PC
Dystrophic	NLS	6.5 ± 0.2	n/d	9.0 ± 0.1	n/d
Eutrophic (with indications of dystrophy)	NLG	n/d	n/d	10.6 ± 0.7	7.8 ± 0.7
Mesotrophic	FQK	10.9 ± 0.2	n/d	4.1 ± 0.6	4.0 ± 0.3
Mesotrophic-eutrophic	DNLB	3.0 ± 0.0	n/d	11.3 ± 0.1	11.4 ± 0.1
Eutrophic	BSC	3.4 ± 0.5	1.0 ± 0.1	6.8 ± 1.4	6.4 ± 0.6
Eutrophic-hypereutrophic	DNLJ	7.3 ± 0.3	n/d	9.7 ± 2.0	10.2 ± 0.1
Hypereutrophic	CLC	9.1 ± 0.2	2.2 ± 0.1	5.4 ± 0.1	1.7 ± 0.2
Hypereutrophic	DFD	9.1 ± 0.7	4.2 ± 0.2	6.4 ± 0.0	n/d

Meaning of abbreviations: SPB—water-suspended particles, collected during *Betula* pollen-spreading period; SPP—water-suspended particles, collected during *Pinus* pollen-spreading period; NLS—natural lake Šermukšnynas; FQK—flooded quarry Kalniškiai; DNLB—dammed natural lake Bijotė; NLG—natural lake Geluva; CLC—surface water of the Curonian Lagoon coast; BSC—surface water of the Baltic Sea coast; DNLJ—dammed natural lake Juodlė; DFD—dammed fishpond Damba;. n/d—no data. The total phenolic content in *Betula* and *Pinus* free PC and bound PC extracts was expressed as rutin equivalents in mg/g of sample dry weight.

**Table 3 plants-13-00099-t003:** Individual phenolic compounds determined by HPLC/DAD in water-suspended particles and surface water.

Phenolic Compounds	PW	SPB	SPBW	SPP	SPPW
*trans*-Ferulic acid	+	+	+	+	+
Vanillic acid	+	+	+	+	+
Gallic acid	+			+	+
Chlorogenic acid			+		
3,4-DHB			+	+	
p-Coumaric acid				+	
Quercetin		+	+		

Meaning of abbreviations: PW—surface water samples, collected before pollen spreading; SPB—water-suspended particles, collected during the *Betula* pollen-spreading period; SPBW—surface water samples, collected during the *Betula* pollen-spreading period; SPP—water-suspended particles, collected during the *Pinus* pollen-spreading period; SPPW—surface water samples, collected during the *Pinus* pollen-spreading period. 3,4-DHB—3,4-dixydroxybenzoic acid. Mobile phase 1: solvent A (100% methanol) and solvent B (10% acetonitrile and 2% acetic acid in water). Mobile phase 2: solvent A (0.01% TFA in water) and solvent B (0.01% TFA in ACN). Flow rate—1 mL/min. Injection volume 20 µL.

**Table 4 plants-13-00099-t004:** Welch’s *t*-test results of total phenolic content in water-suspended particles and surface water according to the trophic status of a water body.

Trophicity	Eutrophic (with Indications of Dystrophy)	Eutrophic	Mesotrophic-Eutrophic	Eutrophic-Hyper-Eutrophic	Hyper-Eutrophic	Mesotrophic	Dystrophic
Eutrophic (with indications of dystrophy)		n/d	n/d	n/d	n/d	n/d	n/d
Eutrophic	5.3 **		1.1	0.2	0.5	1.9	0.5
Mesotrophic-eutrophic	1.9	1.0		1.2	0.9	1.0	1.0
Eutrophic-hypereutrophic	3.1 *	2.3	0.4		0.8	2.0	1.1
Hypereutrophic	7.8 **	0.7	1.4	3.6 *		1.8	0.1
Mesotrophic	8.7 **	1.6	1.9	4.6 **	1.3		1.8
Dystrophic	0.4	3.1*	1.7	1.8	3.6 *	4.1 *	

Absolute *t*-value is shown, which does not indicate the direction of the effect in the comparison between the trophicity. Significant differences at * *p* < 0.05 and ** *p* < 0.01. In the matrix, above the gray-colored cells—*Betula* pollen-spreading period, below the gray-colored cells—*Pinus* pollen-spreading period. n/d—no data. Statistical analysis was performed using data of sample number n = 185.

**Table 5 plants-13-00099-t005:** Main hydromorphological, ecological, and landscape indicators of the studied water bodies [22,24].

Water Bodies and Code	Origin of the Water Body	Water Level Altitude, m	Area of Water Body, km^2^	Mean Depth, m *	Watershed Area, km^2^	Trophic Status	Coastal Landscape **
Lake Geluva, NLG	Natural	134	0.179	3–5	0.61	Eutrophic (with indications of dystrophy)	Forests, swamps
Lake Bijotė, DNLB	Natural (dammed)	128	0.665	4.3	4.32	Mesotrophic–eutrophic	Forests
Lake Juodlė, DNLJ	Natural (dammed)	114	0.340	1–3	2.24	Eutrophic–hypertrophic	Forests, swamps
Lake Šermukšnynas, NLS	Natural	134	0.035	1–3	0.33	Dystrophic	Forests, swamps, bushes
Lake Damba, DFD	Artificial (fishpond)	111	0.545	1–3	1.91	Hypertrophic	Forests, meadows
Lake Kalniškiai, FQK	Artificial (flooded quarry)	99	0.214	1–3	0.35	Mesotrophic	Cultivated fields, meadows, bushes
Curonian Lagoon, CLC	Natural	0	1584	3.8	100,500	Hypertrophic	Forests, swamps, bushes, sandy beaches
Baltic Sea, BSC	Natural	0	412,500	55	2.13 mill.	Eutrophic	Sandy beaches, forests

* The exact average depth is given only for bathymetrically surveyed water bodies. ** The nature of the Curonian Lagoon and Baltic Sea coasts is described only near the sampling sites.

## Data Availability

Data are contained within the article and Appendix A.

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
