# Peer review of "Enrichment of Water Bodies with Phenolic Compounds Released from Betula and Pinus Pollen in Surface Water"

_plants, 2023, doi:10.3390/plants13010099_

Round 1

Reviewer 1 Report

Comments and Suggestions for Authors

The manuscript provides a good study of pollen grains and their spread on water surfaces, along with a study of phenolic compounds and biological activity.

Therefore, the authors succeeded in this manuscript in fully studying this topic.

Comments on the Quality of English Language

Minor editing of the English language required

Author Response

Authors response to Reviewer 1.

The authors express gratitude to the reviewer for their evaluation of the manuscript. We have made several corrections to the English text during the editing process.

Reviewer 2 Report

Comments and Suggestions for Authors

The authors created calibration curves for the standards and re-quantified the phenolic compounds by HPLC. However, there are still two problems, the calibration curves are at concentrations from 0.1 or 0.25 μ/mL up to 1.0 μ/mL (lines 801 to 804). Figure 7 presents a graph with the results in mg/mL(?) and in the text the values are also with the concentration in mg/mL (lines 425, 428, 430...).

Even if they made a mistake in placing mg/mL in figure 7 and in the text, there is still the problem that the maximum concentration value observed (6.6 μ/mL) is far above the maximum concentration used in the calibration curve (1.0 μ/mL). The values measured in the extracts need to be within the concentration ranges used for the calibration curve. It is not correct to extrapolate values using the calibration curve, as points with concentrations above the curve may be in a non-linearity range, which does not follow the Lambert-Beer Law.

In other words, they need to review this curve and the results obtained.

Author Response

Authors response to Reviewer 2.

The authors are very grateful for your valuable comments, observed shortcomings, and constructive suggestions that allowed us to improve the manuscript.

Question. The authors created calibration curves for the standards and re-quantified the phenolic compounds by HPLC. However, there are still two problems, the calibration curves are at concentrations from 0.1 or 0.25 μ/mL up to 1.0 μ/mL (lines 801 to 804). Figure 7 presents a graph with the results in mg/mL(?) and in the text the values are also with the concentration in mg/mL (lines 425, 428, 430...).

Even if they made a mistake in placing mg/mL in figure 7 and in the text, there is still the problem that the maximum concentration value observed (6.6 μ/mL) is far above the maximum concentration used in the calibration curve (1.0 μ/mL). The values measured in the extracts need to be within the concentration ranges used for the calibration curve. It is not correct to extrapolate values using the calibration curve, as points with concentrations above the curve may be in a non-linearity range, which does not follow the Lambert-Beer Law.

In other words, they need to review this curve and the results obtained.

Answer. According to your remarks, the following corrections were made in the manuscript:

Concentrations of individual phenolic compounds in water samples were carefully revised and corrected. All concentrations were expresses as µg/mL.

Calibration curves were built using standards (trans-ferulic, vanillic, gallic, chlorogenic, coumaric, syringic and (di)hydroxybenzoic acids, rutin and quercetin) of 100 µg/mL concentration solutions; and different volumes (1.0, 2.5, 5.0 and 10 µL) of standards were injected. The calculations were done keeping in the mind that real water samples were concentrated from 1L to 5 mL; and injection volumes were 5 µL.

All corrections were indicated in the manuscript text and Supplementary Materials. Also, Figure 7. was revised.

Reviewer 3 Report

Comments and Suggestions for Authors

The research is interesting in the area of the enrichment of water bodies with phenolic compounds released from Betula and Pinus pollen. Nevertheless, the manuscript needs to be improved in order to provide enough information to justify its importance and novelty.

Major comments
Please use the section background to explain the importance and novelty of this research and its objective and re-structure in order to explain the novelty of the research, the research question, hypothesis and the objectives.
Please conclude according to the main objectives of the research and add more practical applicatuion of your study.

Minor comments
Why total polyphenolic compounds was expressed as rutin when in table 3 the gallic acid was present. Please express as gallic acid equivalents.

Author Response

Response to Reviewer 3

The authors express gratitude to the reviewer for their evaluation of the manuscript. We have considered suggestions that improved the quality of the manuscript. We have supplemented the text of the manuscript. Answers to specific questions are given below.

Reviewer. Please use the section background to explain the importance and novelty of this research and its objective and re-structure in order to explain the novelty of the research, the research question, hypothesis and the objectives. Please conclude according to the main objectives of the research and add more practical applicatuion of your study.
Answer. The text of the introduction has been updated to reflect this comment. The novelty, hypothesis and practical utility of the study have been clarified. The revised text is between 97 and 111 lines in length.

Reviewer.  Why total polyphenolic compounds was expressed as rutin when in table 3 the gallic acid was present. Please express as gallic acid equivalents.

Answer.  According to our study, Betula pollen accumulates rutin (Figures 6 and 7) and traces of gallic acid were found. Therefore, we chose to express the total content of phenolic compounds in terms of rutin equivalents. 

Round 2

Reviewer 2 Report

Comments and Suggestions for Authors

I agree with the corrections made to the manuscript.

Reviewer 3 Report

Comments and Suggestions for Authors

Manuscript was improved.